# *Streptococcus salivarius* and *Ligilactobacillus salivarius*: Paragons of Probiotic Potential and Reservoirs of Novel Antimicrobials

**DOI:** 10.3390/microorganisms13030555

**Published:** 2025-02-28

**Authors:** McKinley D. Williams, Leif Smith

**Affiliations:** 1Department of Biology, Texas A&M University, College Station, TX 77843, USA; mwilliams@bio.tamu.edu; 2Antimicrobial Division, Sano Chemicals Inc., Bryan, TX 77808, USA

**Keywords:** *Streptococcus salivarius*, *Ligilactobacillus salivarius*, salivaricins, antibiotic, bacterial drug resistance

## Abstract

This review highlights several basic problems associated with bacterial drug resistance, including the decreasing efficacy of commercially available antimicrobials as well as the related problem of microbiome irregularity and dysbiosis. The article explains that this present situation is addressable through LAB species, such as *Streptococcus salivarius* and *Ligilactobacillus salivarius*, which are well established synthesizers of both broad- and narrow-spectrum antimicrobials. The sheer number of antimicrobials produced by LAB species and the breadth of their biological effects, both in terms of their bacteriostatic/bactericidal abilities and their immunomodulation, make them prime candidates for new probiotics and antibiotics. Given the ease with which several of the molecules can be biochemically engineered and the fact that many of these compounds target evolutionarily constrained target sites, it seems apparent that these compounds and their producing organisms ought to be looked at as the next generation of robust dual action symbiotic drugs.

## 1. Introduction

Disease is an intrinsic staple of the human condition and has been since time immemorial. Within the last century, technological developments have granted us critical insight into the biological causes of disease as well as methods to treat and circumvent illness. Paradoxically, many of these developments have also exacerbated our relationship with disease, and nowhere is this more evident than in the mass usage of antibiotics. Since their introduction at the end of the Second World War, the utilization of antibiotics has become an indelible feature of disease management, particularly within first world nations. The expansive use and availability of these compounds could be safely characterized as a virtual panacea, being responsible for the survival of millions of people who would have otherwise perished due to infection [1,2]. Quite tragically, this same availability has also set the stage for the development of large-scale antibiotic resistance, an increasingly pervasive issue engendered by the systematic abuse and misapplication of these compounds [2]. The overreliance and excessive utilization of antibiotics, both within the agricultural and medical industries, has facilitated the reduced potency of these therapeutics and has led to the promulgation of super-pathogens [3,4,5,6,7]. Examples of these highly aggressive and therapeutically intransigent infections have already been documented, with cases of both antibiotic-resistant gonorrhea and staphylococcal infections showing marked increases in prevalence, associated chronic complications, and lethality [8,9]. This current deficit in available antimicrobials not only constitutes a severe burden on current medical infrastructure but may also pose major risks to global stability, if not dealt with immediately [10].

In addition to the issue of antibiotic resistance, a less obvious problem brought about by antibiotics is their often-deleterious effects on the human microbiota. In recent times, a significant connection has been established between the microbiological composition of the human oral, nasal, and gastrointestinal cavities and the associated health of the host. The correlation between the presence/absence of certain microbial species and the overall degree of species diversity within the microbiota has been shown to increase the onset of several metabolic and autoimmune disorders [11,12,13,14,15,16]. These same irregularities also create greater sensitivity to pathogenic colonization and severe infections from a host of disease-causing agents. Among the ranks of these cellular opportunists include notable respiratory pathogens like SARS-CoV-2, gastrointestinal antagonists like *Clostridium difficile* and *Salmonella*, and cariogenic species like *Streptococcus mutans* [17,18,19,20]. Fascinatingly, this relationship is not just restricted to increased rates of infection but also to neurological features of the animal as well. Termed the gut–brain axis, there is an indelible association between microbial makeup of the host organism and its neurophysiological disposition [20]. While several aspects of the microbiota and brain relationship still require investigation, there is sufficient evidence supporting this basic dynamic and the severe effects that antibiotics might impose on it [21,22,23]. Furthermore, the prevalence of several neurodegenerative and metabolic disorders has increased over the same time period as antibiotic resistance, with several points of evidence suggesting some causal relationship [24,25].

This dilemma of antimicrobial resistance and microbiotic irregularity will require a dynamic method to redress these major problems. To this end, a solution may exist in the probiotic properties of lactic acid producing species, known historically for their eubiotic attributes and for producing several novel antimicrobials [26,27]. *Streptococcus salivarius* and *Ligilactobacillus salivarius* are two such species that fit these criteria, notable for their significant and beneficial contributions to the health of the oral and gastrointestinal microenvironments that they inhabit inside of their human hosts [28,29]. Additionally, the antibacterials produced by these species, often referred as salivaricins, have been demonstrated to exert a protective and stabilizing effect on the host microbiota, attenuating and preventing colonization by more aggressive pathogens and augmenting the host’s immune response [30,31,32,33,34,35].

While the synthesis of antibacterial compounds among commensal species is quite common, LAB species have been identified as major producers of several unique subsets, including the structurally dynamic lantibiotics produced by *S. salivarius*. The lantibiotic class of antimicrobials is of particular interest in this regard, given its many interesting features, including its broad spectrum of activity, structural modularity, amenability to chemical modification, structural and biosynthetic diversity, and its several modes of action involving a highly conserved target site [36,37,38,39,40,41,42,43,44]. Furthermore, some of these compounds have demonstrated a critical role in probiotic immunomodulation and species composition within humans, a property that they likely evolved due to their commensal association with *Homo sapiens* [39,45].

## 2. *Streptococcus salivarius* and Its Bacteriocins

*S. salivarius* is a Gram-positive bacterium that is thought to be one of the earliest species to colonize the human microbiota and is localized predominantly in the mouth and, to a lesser extent, in the upper respiratory system [46]. This same species is also noted for its apparent immunomodulatory capacities within humans [47]. One nexus between *S. salivarius*’ presence within the oral microbiota and its apparent effects on the host immune response may be found in the species’ production of several effective antimicrobials. Indeed, the presence of these compounds in the environment is not inconsequential, with research showing that they are critical for helping to regulate the microbial make-up of the human oralome [2,48,49]. This observation would indicate a potential for these molecules to be employed as alternative antibiotics, a proposal merited by the apparent effectiveness of these compounds against pathogenic streptococcal species, including *Streptococcus pyogenes* and *Streptococcus agalactiae* [2,48,50]. Further validation of this point can be found in the long-standing market viability of the K12 strain of *S. salivarius*, a commercially available probiotic used in the treatment of halitosis [2,51].

Salivaricin is a general descriptor for all varieties of compounds produced by either *S. salivarius* or the distantly related *Ligilactobacillus salivarius,* which have demonstrable bactericidal/bacteriostatic properties (Table 1). These compounds are ribosomally synthesized and can exhibit either broad-spectrum toxicity or acute target specificity [2,52]. These same molecules can be divided and subdivided in accordance with their molecular weight, chemical composition, biosynthesis, and mode of action. More precisely, these bacteriocins can be catalogued into one of four groups: (I) lantibiotics, (II) small peptides, (III) large peptides, and (IV) cyclized [2,53]. Groups II and III are noted for their lack of post-translational modifications (PTMs) and are easily distinguished from one another by whether they fall above or below a 10 kD threshold. Group IV bacteriocins have only been recently identified and, consequently, are the least understood of all bacteriocin classes. They are characterized by the presence of PTMs involving cyclization and addition of carbohydrate and lipid moieties. The lantibiotics and their many structural subclasses are some of the best studied bacteriocins to date. For this reason, they will be the primary focus of this review, with particular interest directed towards the lantibiotic salivaricins, henceforth referred to as lantho-salivaricins within the present article.

## 3. Biosynthesis and Bioactivity of the Lantibiotic salivaricins

Much of what is known about lantho-salivaricin biosynthesis comes from what has been learned in other bacterial systems that produce similar compounds. Lantibiotics are characterized by a number of PTMs, the most notable of these being the lanthionine rings. These rings are produced through a set of dehydration reactions that convert serine/threonine amino acids into dehydroalanine/dehydrobutyrine residues, which can then form thioether linkages with cysteines [2,63]. Typically, the formation of these linkages is achieved enzymatically through cyclases, proteins responsible for the efficient generation and positioning of covalent bonds between the dehydro-amino acids and the thiol group of cysteines [2,64]. It should be noted that non-catalyzed cyclizations are possible, often as a result of spontaneous stereoselective Michael additions that occur between key substrates [2,65]. While this is certainly one mode by which these modifications are generated, such spontaneous formations of lanthionine are not the conventional way in which such structures are produced in functional lantibiotics.

Regarding the salivaricins’ modes of action, it is important to note that the specific mechanisms by which they achieve their bacteriostatic/bactericidal responses are by no means uniform. These molecules exhibit several diverse mechanisms to achieve their antimicrobial effects which include, but are not limited to, pore formation, lipid II sequestration, and growth inhibition [2,52,66,67]. This variation in activity is attributable to these molecules’ structural diversity, which often informs and affects the particular manner in which they interact with their targets [2,68]. In the case of lantibiotics, the lanthionine rings, as illustrated in Figure 1, serve as the primary source of bioactivity and chemical stability. While examples to the contrary exist, lantibiotics generally work via direct binding to the cell wall precursor lipid II [2,63]. When this type of bacteriocin complexes with the target substrate, it undergoes a conformational change wherein a direct interaction is formed between the lantibiotic and the pyrophosphate group of the target, resulting in the formation of a cage-like heterodimer [2,69,70]. The consequences of such an interaction can be quite variable in nature. In some instances, these interactions can result in inhibitory responses that only retard growth in the target organism. In other scenarios, the interfacing of lipid II with specific lantibiotics can induce more problematic disruptions, including perturbations of membrane potential, efflux of vital salts, influx of cytotoxic compounds, loss of ATP, and cell death [2,71]. Regarding the lantho-salivaricins specifically, modes of action have been shown to be dynamic, with mechanisms including apparent interference with cell wall formation, as is the case for salivaricin B, and possible membrane perforation, as is predicted to be the case for the nisin-like salivaricin D [55,72].

### 3.1. Type 1 Lantho-Salivaricins

Similar to bacteriocins more broadly, the lantibiotics can be categorized and subcategorized based upon different criteria, including the specific enzymes involved in their biosynthesis as well as the presence of certain PTMs [2,73]. For this review, the classification system first proposed by Wiley and van der Donk (2007) [2,74] will be used and discussion will be restricted to the first two sub-classes (hereafter referred to as types, to avoid confusion). Type I and type II lantibiotics are distinguished by the biosynthetic mechanisms involved in the generation of their ring motifs after translation of the prepeptide. For type I, the dehydration of the serine/threonine amino acids is facilitated through the LanC dehydratase that deprotonates the amino acids via glutamylation [2,44,75,76]. The dehydroalanine/dehydrobutyrine residues generated from this reaction participate in thioether linkages with cysteine residues. These formations are achieved via LanC, which is needed for the site-specific coupling of thiol-bearing cysteines to dehydro-amino acids [2,77]. The only type I lantibiotic salivaricin that has been identified thus far is salivaricin D [72].

### 3.2. Type II Lantho-Salivaricins

For type IIs, the dehydration of the serine/threonine amino acids is achieved through a dual phosphorylation/elimination reaction, whereby the introduction and subsequent removal of phosphates is catalyzed by specific sub-domains within the effector enzyme [2,78,79,80]. Unlike type I lantibiotics that have distinct enzymes for the dehydration and cyclization steps, structural maturation of class II lantibiotics is facilitated via a single bifunctional synthase, LanM, responsible for the installation of both the dehydro-amino acids and thioether bridges [81,82]. Type II lantibiotics can be further subdivided into single-component and two-component varieties. The list of type II lantibiotics is extensive, including lacticin 481 as well those designated as salivaricins, such as salivaricin A(2), salivaricin B, salivaricin E, salivaricin G32, salivaricin 9, and salivaricin 10 [44,46,54,56,57,83].

### 3.3. Two-Component Salivaricins

Two-component bacteriocins can be either lantibiotics or belong to the non-modified classes of antimicrobials (class II and III). As their name would suggest, two-component bacteriocins consist of two distinct precursor peptides that must undergo processing and structural maturation before acquiring synergistic activity [2,84]. The synergistic nature of this sub-type notwithstanding, two-component lantibiotics, such as lacticin 3147, possess more unique structural augmentations in the form of D-alanine amino acids, PTMs that are generated after the synthesis of the prepeptide via the reductase/dehydrogenase, LtnJ [2,85,86,87]. Besides some of the compound-specific tailoring modifications, much of the biosynthesis of these two-component lantibiotics is identical to other type IIs except that two LanM proteins are involved in the synthesis of each of the components [88]. As of yet, no examples of two-component lantho-salivaricins have been identified. Instead, the two-component salivaricins that have been documented, have all been of the non-modified variety produced by *L. salivarius*, including salivaricin P, salivaricin T, salivaricin CRL 1328, and salivaricin APB-118 [58,59,60,62].

### 3.4. Mode of Action of Salivaricins

The number, position, and topology of lanthionine rings, along with the proper modality of key binding motifs, are vital for optimal target specificity among the lantibiotics [2,89,90]. Additionally, these same ring motifs are essential for the native thermostatic and proteolytic stability of these peptides [2,91,92]. Depending upon the nature of these structures, certain bactericidal responses can be seen. In the case of gallidermin, both cidal and static mechanisms have been observed. When gallidermin binds to lipid II in the target, it will block cell wall synthesis while simultaneously inducing pore formation [2,93]. The effectiveness of this two-pronged attack appears to be dependent upon the thickness of the target’s membrane [2,93]. Among the lantho-salivaricins, lipid II is commonly accepted as the target moiety; however, in many instances, the specific manner by which these compounds achieve their inhibitory responses has not been experimentally elucidated. It has been presumed that compounds such as salivaricin G32 facilitate their bactericidal responses via pore formation, based upon their structural similarities to other, better studied pore forming lantibiotics like SA-FF22 [57,94]. By this same metric, it has been proposed that salivaricin D possesses a similar mode of action to its structural homolog nisin, a well-studied lantibiotic that is known for both its induction of pore formation and its inhibition of cell wall synthesis via lipid II septum-abduction [55,95,96]. It should be noted that while it may be convenient to extrapolate protein function through homology alone, such enterprises are known to have severe limitations and be quite unreliable when determining the true functional capacity of a molecule [97]. For this reason, experimental determination and verification is important when appreciating the effect of these molecules, especially for the purpose of fully exploiting their abilities as potential antimicrobials.

In this context, there are a handful of lantho-salivaricins with experimentally defined modes of actions. In the case of salivaricin 9, its mechanism appears to be identical to several conventional non-salivaricin lantibiotics like subtilin and epidermin, causing perforation of the cell membrane [98,99]. By contrast, the lantibiotic salivaricin B was demonstrated to have a mode of action similar to that of vancomycin, inducing cell death via the blocking of transglycosylation between lipid II subunits [72]. This mode of inhibition was shown to take place without the introduction of any discernable damage to the cell membrane. Modification experiments with salivaricin A(2) revealed that the presumably unstructured N-terminal region is critical for both lipid II binding and overall bioactivity of the compound [44]. Several N-terminal semisynthetic variants of salivaricin A(2) were synthesized demonstrating the importance of the N-terminal Lys–Arg residues for bioactivity. While these observations alone might suggest a similar mechanism of action to salivaricin 9 or salivaricin B, it was noted that salivaricin A(2) possessed inferior binding affinity when compared to other standard lantibiotics, suggesting that its particular mechanism may be distinct [44].

In the case of two-component lantibiotics, the modes of action are generally identical to one-component varieties—through lipid II binding and introducing membrane perforation and/or blocking transglycosylation [99,100]. Like nisin for type I lantibiotics, lacticin 3147 is one of the best characterized of the two-component lantibiotics in regards to its structure, homology, and its mechanism for cell–membrane interaction [86,101,102,103,104,105]. A robust model for these compounds’ mode of action has been established.

For two-component lantibiotics, the α and β-units work together to permeabilize the cell membrane of the target [2,84,106]. The α-unit of these compounds will initiate this process through direct attachment to lipid II, after which the β-unit will complex with the lipid II heterodimer, allowing for pore formation to occur [2,106,107]. Curiously, the high-affinity binding characteristic of the α-unit is due to it possessing a binding region for lipid II that is structurally similar to mersacidin, a single-component class II lantibiotic [2,108]. However, it does appear that activity of these two-component molecules is generally most optimal under conditions where both constituents are present in equimolar concentrations [2,101,109,110]. The synergy between the subunits appears to be dependent upon the presence of lipid II and its accessibility to the α-unit. This is evidenced by the fact that, when β-units are added to a target before α-units, the holopeptides do not form and no activity is observed [2,101]. Based upon these observations, it was proposed that these compounds synergize via a three-step process: (I) association of the α-unit with lipid II, (II) subsequent conformational change of complex enabling docking of the β-unit, and (III) docking of the β-unit and formation of a trimer capable of introducing pores into the target membrane [2,101,111]. Several two-component lantibiotics, including lacticin 3147, haloduracin, and thusin, have been shown to have activities consistent with this model [2,110,111,112,113].

It should be stated for purposes of clarity that class II bacteriocins (non-modified) produced by *L. salivarius* do not typically utilize lipid II as a critical binding site for bioactivity. While a specific binding site for class II salivaricins has not yet been identified, their probable mode of action appears to be through pore formation, as evidenced by experiments with salivaricin mmaye1 [61]. Much of the work related to binding activities and mechanisms of action for class II bacteriocins is derived from related molecules, wherein most of these compounds also facilitate cell death through pore formation [114,115]. Alternatively, some class II bacteriocins, such as lactococcin 972, inhibit growth via lipid II sequestration (the only class II bacteriocin known to do this), while others, such as mesentericin Y105, can introduce membrane pores but may also act as virulence factors, based upon their ability to induce mitochondrial uncoupling [116,117,118]. With respect to pore formation, class IIa bacteriocins achieve this through their N terminals engaging in electrostatic binding with the IIC/IID subunits of the mannose phosphotransferase [119,120,121]. This binding of the N-terminus then allows for the integration of the C-terminal domain into the interior of the transport protein, inducing an irreversible change in the transporter’s conformation and resulting in a loss of membrane potential [119,121,122].

## 4. Importance of Salivaricin-Producers in the Human Microbiota

The human microbiota and its relationship with the organismal self are both intricate and fascinating aspects of nature. While still enigmatic in many respects, research into the microbiota is an important facet of biological study, particularly in relation to personal health. The full extent of these interactions between the host and the microbiota is not fully known. What is understood is that the confluence of interactions between the host and its microbiota have created an essential homeostatic balance that must be maintained to ensure organismal stability [123,124]. Whether due to the use/misuse of antibiotics, exposure to toxins, disease, or sub-optimal diets rich in polysaccharides and fats, actions that serve to subvert this balance and induce dysbiosis carry negative health implications [125,126,127,128]. Consequences associated with an inability to maintain this balance are numerous, but generally involve the onset of various opportunistic infections, inflammation, and both metabolic and immunological disruptions [123]. It is for these reasons that much research has been directed toward the identification and development of probiotic species that can help to supplement host health and maintain stability. Besides being a wellspring of antimicrobials, there is strong evidence to show that *S. salivarius* may serve as a sufficient probiotic, particularly for the upper respiratory system and mouth.

While highly competitive in nature, the human oral cavity is a niche that *S. salivarius* efficiently occupies whilst simultaneously exerting a significant and beneficial influence over the host immune response [2,47,129,130]. It must be stressed that the mouth and its vast web of complex symbiotic associations are quite intricate, involving a diverse array of microbial inhabitants, including bacteria, phage, and fungi [2,131]. Furthermore, the species that colonize the mouth are highly resistant to fluctuations in organismal composition and appear to be resistant to outside colonization [2,132,133]. Much like the wider microbiota, the host of interactions occurring in this locality are not comprehensively understood, but what is recognized is the generalized stability of the oralome.

Much of the research pertaining to ecological stability and oral health appears to implicate several different species of *Streptococci* bacteria, demonstrating their importance in either suppressing or managing invasive and/or pathogenic elements [2,133,134,135]. Indeed, many species, including *S. salivarius* and other LAB, have proven to have strong stabilizing effects in the face of dysbiosis and antibiotic-resistant colonizers, including those closely associated with the GI tract [6,136,137]. Furthermore, many of these same compounds and organisms also have positive effects on the immune system, working synergistically with adaptive and innate components to confer a robust defense against colonizing agents, whether localized in the gut or in the upper respiratory tract [129,138,139].

*S. salivarius* has many very specific means by which it serves as a probiotic, in addition to the production of bacteriocins and similar antimicrobials. Some strains of *S. salivarius* are able to achieve a reduction in antagonistic colonization via competitive exclusion. Active strains are known to exhibit binding affinity to the epithelial wall of the mucosa, the same binding site that is required for many pathogenic species [139,140,141]. This has the obvious effect of reducing successful colonization on the part of the antagonist. Other probiotic inducements by *S. salivarius* involve causing changes in the host’s immune response [142,143]. Compounds generated by these species are known to have effects that include, but not are not limited to, modulation of pro-inflammatory cytokines, inhibition of NF-kB expression, reduction in transcriptional activity of PPARy, and suppression of arthritis [47,129,144,145,146,147]. One example of this is how *S. salivarius* works synergistically with other bacterial species to induce increases in macrophage responsiveness through promoting the upregulation of TNF-α [148]. It has also demonstrated an ability to affect the inflammation response of host epithelium by acting on specific metabolic genes. Specifically, *S. salivarius* can induce transcriptional downregulation of PPARγ, a transcription factor involved in various developmental, metabolic, and immunological processes, including but not limited to tissue differentiation, glucose homeostasis, and anti-inflammatory responses [144,149]. Further probiotic inductions are achievable by *S. salivarius* through its ability to modulate the behavior of the host microbiome in ways that are highly beneficial to metabolic health, particularly in areas associated with oral eubiosis [45,150].

Similarly to *S. salivarius*, *L. salivarius* has a very intimate association with the human host, serving as a major probiotic. Interestingly, *L. salivarius* has a much more expansive presence in the human host in comparison to *S. salivarius*, being found throughout a vast range of tissues including the GI tract, the upper respiratory system, and the vaginal cavity [151]. Across this larger domain, *L. salivarius* is able to engage in various immunomodulator augmentations. Some of these effects include the ability to enhance macrophage activity, modulation of anti-inflammatory cytokines such as IL-10, downregulation of TNF-α expression, competitive exclusion of pathogenic species from host tissues, and the increase in the number of Th1 cells [152,153,154].

In addition to these responses, *L. salivarius* has also demonstrated significant therapeutic abilities in the amelioration of specific dysbiotic conditions associated with the vaginal cavity. More precisely, experiments with *L. salivarius* cultures showed them to be highly effective in clearing bacterial vaginosis, whilst they were simultaneously able to induce a corresponding reduction in pro-inflammatory vaginal cytokines, such as IL-1β and IL-6 [155,156,157]. These positive effects also extend into the realm of improved fecundity in women who were attempting pregnancy through IVF or who suffered from either recurrent pregnancy loss or general infertility [158,159]. Studies demonstrated that women who had improved fecundity after consistent application of *L. salivarius* culture also experienced increased levels of TGF-β1, a growth factor critical for maintaining fetal–maternal immune tolerance and, by extension, a successful pregnancy [159,160].

## 5. Phosphorylated Salivaricin

More recently, a novel phospho-modification was identified in the lantibiotic bacteriocin, salivaricin 10 [45]. Synthesized by the sal10 strain of *S. salivarius*, salivaricin 10 is a class II lantibiotic that was found to have a phosphothreonine at position 4 along its N-terminus (the first of any lantibiotic known to possess such a PTM). While possessing a similar bioactivity spectrum to other reported lantibiotics, it was also found to have unique immunomodulatory capabilities that were, in part, due to the presence of the phosphate moiety. It was noted that the immunomodulation of the compound was two-pronged in nature. In one respect, salivaricin 10 was shown to directly augment host immunity through activation of neutrophils, downregulation of pro-inflammatory macrophages, and upregulation of anti-inflammatory macrophages. Secondly, the compound demonstrated highly selective activity, antagonizing species associated with cariogenicity and malodorousness, such as *S. mutans* and *Bifidobacterium dentium*, while not affecting known oral commensals and eubiotic species like *Streptococcus parasanguinis*. The kind of broad species selectivity exhibited by salivaricin 10 is quite unique, compared to the more generalized activity observed in lantibiotics such as nisin. In this manner, salivaricin 10 modulates the immune response of the host by shifting the species composition of the oral cavity from a largely inflammatory dysbiotic state to a eubiotic one [45].

## 6. Conclusions and Perspectives

Rapid globalization and market integration of both agricultural and medical industries have reconfigured much of the international landscape, and this has invariably offered greater innovation and technological accessibility to people throughout the world. Unfortunately, this same wide-scale integration of markets, medicine, food, and international cooperation has also set the stage for a world that, while much wealthier, may, in some respects, find itself in a much less enviable state then it did in the time prior to the end of the Second World War.

To be precise, the globalized economy has created several major and interconnected problems that are setting the stage for disaster. The magnitude of technological development at the turn of last century, paired with the unprecedented level of global peace achieved at the end of the Second World War, has led to a marked increase in the global population. While this development has led to an increase in human capital, it has also led to the inescapable need for greater food production. Consequently, the population of livestock has increased by orders of magnitude, often being reared under cramped and unhygienic conditions conducive to the spread of disease, thus necessitating the use of antibiotics. This in turn has accelerated the evolution of more resistant strains of bacterial species, most notably seen in resistant zoonotic strains of *Staphylococcus aureus,* which have made their way into the human population via this large-scale factory farming [161,162].

The cavalier nature with which antibiotics are issued to livestock is also paralleled in our ever-growing human population, wherein significant abuse and misuse of these same compounds is rife. This same marked increase in the global population has also incentivized measures to provide cheaper sources of calories, usually in the form of highly processed carbohydrates that are known to exert a deleterious effect on the microbiome. Taken together, our abundant and highly industrialized world may not only lead us back into a pre-antibiotic age, but into one characterized by a population that has a comparatively worse constitution then its forebearers: it will be much sicker, metabolically unwell, and susceptible to increasingly formidable pathogens, and all without the assistance of antibiotics. It is apparent in our current age of waning antimicrobial efficacy and increasing gut dysbiosis that new dynamic therapeutics must be developed.

The onset of increased antibiotic resistance and disease is a daunting and potentially cataclysmic scenario that must be addressed in short order. In light of this, we might take consolation in the existence of the various probiotic species, chief among them being *S. salivarius* and *L. salivarius*, which provide several biological benefits that can be readily exploited. As detailed earlier in the review, both LAB species possess several biological properties that can drastically alter the immune system through their presence, and can also change the species composition of the niche through the production of various bacteriocins. These compounds, particularly the lantibiotics, have consistently demonstrated significant broad-spectrum activity against several species of known pathogenic agents. This wide spectrum of activity is attributable to their substantial binding affinity to evolutionarily constrained molecules, insuring a much longer period of commercial viability, if utilized as an antibiotic. Precedents have already been established in using streptococcal derived compounds as probiotic immunomodulators, as reflected in the decades-long application and market viability of salivarius K12. Furthermore, the lantibiotics produced by this species are also readily amenable to biochemical modification, which has significantly enabled improved production and bioactivity of these compounds.

The discovery that salivaricin 10 mediates immunomodulation through both signal transduction of immune cells and selective bactericidal activity should further highlight the therapeutic potential of these compounds. If the probiotic and antibacterial functions observed in this species and its associated compounds are both as modular and interconnected as they appear to be, we might consider this the foundation for symbiotic drugs. In other words, effective use of these probiotics/bacteriocins may entail combining variable dosages of pharmaceutical grade bacteriocins with LAB probiotics, so as to optimize the therapeutic potential of both.

While an equivalent probiotic effector has not been experimentally identified in *L. salivarius*, it seems probable that such molecules play a role in its immunomodulatory operations, given the presence of effector molecules in the closely related *Lactobacillus plantarum*, which is known to produce pyroglutamic acid dipeptides that actively lower expression levels of IFN-γ [163,164]. It seems evident that, through careful study and bioengineering, it may be possible for these compounds to serve a necessary role as the next generation of robust, symbiotic therapeutics; acting as dual anti/prebiotics that will be essential for preventing a post-antibiotic future.

## Figures and Tables

**Figure 1 microorganisms-13-00555-f001:**
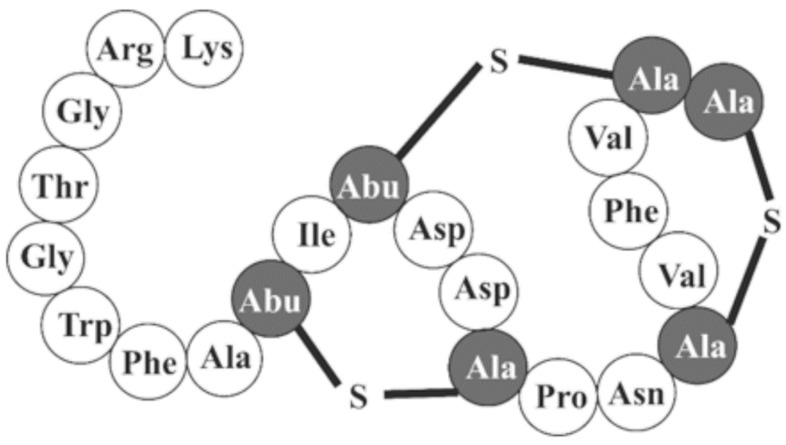
Confirmed structure of lantibiotic salivaricin A(2) including the lanthionine rings and dehydro-amino acids characteristic of all class I bacteriocins. Lantho-salivaricins are all structurally similar to one another regarding their PTMs; however, some compound specific variations are present.

**Table 1 microorganisms-13-00555-t001:** Identified salivaricins and their associated biological, structural, and functional characteristics.

Salivaricin Name	Bacteriocin Class	Producer Strain	Mode of Action	Highest Level of Identification/Structural Characterization	Reference
Salivaricin A(2)	Class I: Lantibiotic ^II^	*S. salivarius* K12	N-terminal Lipid II Binding	NMR and Mass Spectrometry	[44]
Salivaricin B	Class I: Lantibiotic ^II^	*S. salivarius* K12	Inhibition of Transglycosylation	Genomic Prediction	[54]
Salivaricin D	Class 1: Lantibiotic ^I^	*S. salivarius* 5M6C	Unknown (Suspected to be pore formation)	Genomic Prediction	[55]
Salivaricin E	Class I: Lantibiotic ^II^	*S. salivarius* JH	Unknown	Genomic Prediction	[56]
Salivaricin G32	Class 1: Lantibiotic ^II^	*S. salivarius* G32	Unknown (Suspected to be pore formation)	Genomic Prediction	[57]
Salivaricin 9	Class I: Lantibiotic ^II^	*S. salivarius* 9	Pore Formation	Genomic Prediction	[51]
Salivaricin 10	Class I: Lantibiotic ^II^	*S. salivarius* SALI10	Untested (possesses anti-biofilm activities)	NMR and Mass Spectrometry	[45]
Salivaricin L	Class II	*L. salivarius* DPC 6488	Untested	Genomic Prediction	[58]
Salivaricin P	Class II	*L. salivarius* DPC 6005	Untested	Genomic Prediction	[59]
Salivaricin T	Class II	*L. salivarius* DPC 6488	Untested	Genomic Prediction	[58]
Salivaricin APB-118	Class II	*L. salivarius* UCC118	Pore Formation	Genomic Prediction	[60]
Salivaricin mmaye1	Class II	*L. salivarius* SPW1	Pore Formation	Chromatographic Isolation and Mass Spectrometry	[61]
Salivaricin CRL 1328	Class II	*L. salivarius* CRL 1328	Pore Formation	Genomic Prediction	[62]

^I^ Denotes the Type I lantibiotic subclass. ^II^ Denotes the Type II lantibiotic subclass.

## Data Availability

No new data were created or analyzed in this study.

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
