# Peer review of "Streptococcus salivarius and Ligilactobacillus salivarius: Paragons of Probiotic Potential and Reservoirs of Novel Antimicrobials"

_microorganisms, 2025, doi:10.3390/microorganisms13030555_

Round 1

Reviewer 1 Report

Comments and Suggestions for Authors

The review article entitles “Streptococcus salivarius and Ligilactobacillus salivarius: A Paragon of Probiotic Potential and Reservoir of Novel Antimicrobials” is quite interesting and discuss a topic of current needs. As a whole, the review is well written; some parts need to be added with more information below.

  1. The introductory portion is well written, although it needs to be provided with more information to easily understand the main concept. The importance of probiotics and human microbiota metabolites should be added; also the two main bacteria that is the centre of this review should be explained a little (Streptococcus salivarius and Ligilactobacillus salivarius), The authors can cite the following manuscript if they find it useful.

https://doi.org/10.1080/19490976.2024.2431643

https://doi.org/10.1039/D4NP00042K

  1. Line 140-144, in which case it is an inhibitory response, rearrange this sentence to understand easily.
  2. The section 4 should be added with more information regarding the importance of metabolites of target bacteria, and how they can contribute to different direction.
  3. In conclusion, give a little description to probiotics also.
  4. There is much plagiarism between line 232-254, please rearrange it
  5. As a suggestion, one figure can be added to show the importance of these two bacteria as probiotic and antimicrobials

Author Response

Responses to reviewer 1:

Comment 1: Further information has been added to intro to help in communicating the primary point of the article. The two suggested references have also been incorporated into the manuscript.

Comment 2: Line 140-141 has been restructured to improve comprehensibility

Comment 3: This is a fair observation; however it should be noted that very little is known about specific ways that bacteriocins directly mediate immunomodulatory changes. Barbour et. al, 2023 (Reference 45 which is cited extensively in section 5) was significant because it was one of the first times a group was able to provide evidence for a salivaricin bacteriocin to directly mediating an immune response.

Comment 4: The conclusion has been modified and expanded to give more detail about the probiotic aspect of the organisms.

Comment 5: Much of the language in the present manuscript has been adapted from the primary author’s [McKinley Williams] master’s thesis [reference 2 in the manuscript]. The “plagiarism” in lines 232-254 is an artifact of this usage. The primary author is the sole copyright holder of the text in which the language is borrowed from and we have permission to reproduce the text from Texas A&M University (archival institution) under the condition that the thesis is cited. We do have official correspondence from the university to confirm these statements and are more than happy to supply them if requested. Given all this, we do not feel it necessary to drastically alter or rearrange the text as it is currently presented in the manuscript. With that said we do of course respect these concerns and will be happy to comply with a request to adjust the text if it is thought necessary by the editor. 

Comment 6: While this suggestion is understandable, we feel that a figure may not thoroughly communicate the more complex aspects of these organism and their antibacterial/probiotic properties. The list of citations provided in the manuscript are felt to be much better at communicating the nuances.  

Reviewer 2 Report

Comments and Suggestions for Authors

The review submitted by Williams and Smith is highly interesting and provides an in-depth description of the mechanisms of action of salivaricins, along with practical application examples. The manuscript is robust and well-structured, summarizing important information from numerous references.

The only aspect that I found somewhat superficial is the discussion on the practical applications of salivaricins and their cost-benefit comparison with conventional antibiotics. If such information is available, I suggest its inclusion to enhance the applied relevance of the review.

Minor Remarks:

  • References throughout the text need to be adjusted to match the journal’s style.
  • The title of Table 1 should be improved to make it self-explanatory.
  • Keywords: Avoid using words that are already in the title. Replace them with other terms that better contextualize the study for improved article indexing.
  • Line 42: The phrase "Staph infections" is not scientifically accurate. Please rephrase appropriately.
  • Lines 48, 89, and several other instances: Avoid using the term floral, as it refers to flowers. Use microbiota consistently throughout the manuscript.
  • Line 69: Do not abbreviate the genus name unless it has been previously written in full.
  • Table 1: The formatting needs improvement, as it currently appears more like a figure than a properly structured table. Adjust it to match the journal’s formatting style.

Author Response

Responses to reviewer 2:

Primary Comment: Discussion on the cost-benefit comparison is indeed a very interesting suggestion by the reviewer. With that said, there is not a sufficient amount of published data that would invite us to offer well-informed commentary on the topic.

Comments on Reference and Table: In-text citations have been adjusted to bring them into line with the formatting of the journal. The title for table 1 was has been adjusted to offer more clarity to the information presented.

Comment on Keywords: We have made adjustments to the language in various lines throughout the manuscript to reduce the use of specific words used in the title

Comment on Line 42: Issue has been addressed in the manuscript

Comment on Line 48, 89, etc: Floral has been replaced with more appropriate language describing the microbiota and microbial inhabitants.

Comment on Line 69: Issue has been addressed in the manuscript

Comment on Table 1: Further clarification may be required here. Guidelines provided in “Preparing Figures, Schemes, and Tables” in the “Instructions for Authors” page on the MDPI website are somewhat broad. It does state that “All table columns should have an explanatory heading... Authors should use the “Table” option in Microsoft Word to create tables”. Our table does not appear irregular when compared to other tables in similar publications in your journal. More specifics will be greatly appreciated if there is an oversight on our part.  

Round 2

Reviewer 1 Report

Comments and Suggestions for Authors

The manuscript has been revised well.